# A General Protocol to Probe Large Vision Models for 3D Physical Understanding

**Guanqi Zhan**
VGG, University of Oxford
guanqi@robots.ox.ac.uk

**Chuanxia Zheng**
VGG, University of Oxford
cxzheng@robots.ox.ac.uk

**Weidi Xie**
VGG, University of Oxford
SAI, Shanghai Jiao Tong University
weidi@robots.ox.ac.uk

**Andrew Zisserman**
VGG, University of Oxford
az@robots.ox.ac.uk

## Abstract

Our objective in this paper is to probe large vision models to determine to what extent they 'understand' different physical properties of the 3D scene depicted in an image. To this end, we make the following contributions: (i) We introduce a *general* and *lightweight* protocol to evaluate whether features of an off-the-shelf large vision model encode a number of physical 'properties' of the 3D scene, by training discriminative classifiers on the features for these properties. The probes are applied on datasets of real images with annotations for the property. (ii) We apply this protocol to properties covering scene geometry, scene material, support relations, lighting, and view-dependent measures, and large vision models including CLIP, DINOv1, DINOv2, VQGAN, Stable Diffusion. (iii) We find that features from Stable Diffusion and DINOv2 are good for discriminative learning of a number of properties, including scene geometry, support relations, shadows and depth, but less performant for occlusion and material, while outperforming DINOv1, CLIP and VQGAN for all properties. (iv) It is observed that different time steps of Stable Diffusion features, as well as different transformer layers of DINO/CLIP/VQGAN, are good at different properties, unlocking potential applications of 3D physical understanding. Our project page is https://www.robots.ox.ac.uk/~vgg/research/phy-sd/.

## 1 Introduction

The large-scale pre-trained vision foundation models have achieved great success in computer vision tasks, including classification (CLIP [18, 30]), segmentation (DINO [5, 27]), and image generation (VQGAN [11], Stable Diffusion [31]) with strong generalisation capabilities. However, they are mainly trained with 2D images, which are the projection of the 3D physical world. This naturally raises the question of to what extent these large-scale vision models have learned about the 3D scene depicted with only the 2D images. Our objective in this paper is to investigate this question, and we do this precisely by determining whether features from these large-scale pre-trained vision models can be used to estimate the true geometric and physical properties of the 3D scene. If they can, then that is evidence that the models encode the 3D properties. For example, as an indication that Stable Diffusion is 3D and physics aware, Figure 1 shows the result of the off-the-shelf Stable Diffusion model [31] inpainting masked regions in real images – it correctly predicts shadows and supporting structures.

38th Conference on Neural Information Processing Systems (NeurIPS 2024).

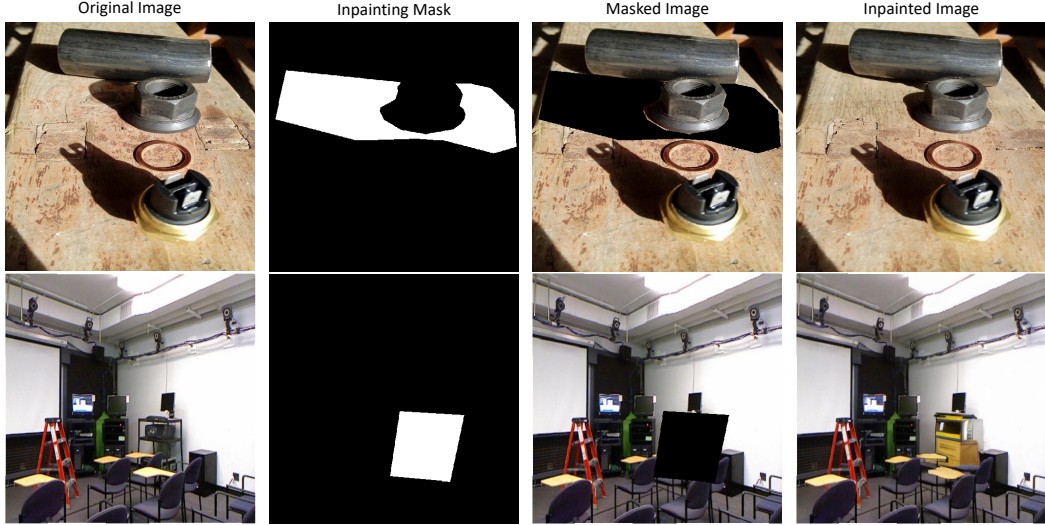

| Original Image | Inpainting Mask | Masked Image | Inpainted Image |

Figure 1: **Motivation: What do large vision models know about the 3D scene?** We take Stable Diffusion as an example because Stable Diffusion is generative, and so its output is an image that can be judged directly for verisimilitude. The Stable Diffusion inpainting model is here tasked with inpainting the masked region of the real images. It correctly predicts a shadow consistent with the lighting direction (top), and a supporting structure consistent with the scene geometry (bottom). This indicates that the Stable Diffusion model generation is consistent with the geometry (of the light source direction) and physical (support) properties. These examples are only for illustration and we probe a general Stable Diffusion network to determine whether there are explicit features for such 3D scene properties. The appendix provides more examples of Stable Diffusion's capability to predict different physical properties of the scene.

To answer this question, we propose a *general* and *lightweight* evaluation protocol to *systematically* and *efficiently* 'probe' a pre-trained network on its ability to represent a number of 'properties' of the 3D scene and viewpoint. The protocol could be used for any network and any property of interest. We have probed properties including: 3D structure and material of the scene, such as surface layout; lighting, such as object-shadow relationships; and viewpoint dependent relations such as occlusion and depth.

The protocol involves three steps: *First*, a suitable real image evaluation dataset is selected that contains ground truth annotations for the property of interest, for example the SOBA dataset [41] is used to probe the understanding of lighting, as it has annotations for object-shadow associations. This provides a train/val/test set for that property; *Second*, a grid search is carried out over the layers and time steps of the Stable Diffusion model, and transformer layers for other models, to select the optimal feature for determining that property. The selection involves learning the weights of a simple linear classifier for that property (*e.g.* 'are these two regions in an object-shadow relationship or not'); *Third*, the selected feature (layer, time step) and trained classifier are evaluated on a test set, and its performance answers the question of how well the model 'understands' that property.

In short, we probe scene geometry, material, support relation, shadow, occlusion and depth, to answer the question "To what extent do large vision models encode 3D properties of the scene?". We apply this protocol to a wide range of networks that are trained at large scale, including OpenCLIP [18, 30], DINOv1 [5], DINOv2 [27], and VQGAN [11]. This covers networks trained generatively (Stable Diffusion), with self-supervision (DINOv1 & DINOv2), with weak supervision (OpenCLIP), and by auto-regression (VQGAN).

From our investigation, we make three observations: *First*, the Stable Diffusion and DINOv2 networks have a good 'understanding' of the scene geometry, support relations, the lighting, and the depth of a scene, with Stable Diffusion and DINOv2 having a similar and high prediction performance for these properties. However, their prediction of material and occlusion is poorer. *Second*, Stable Diffusion and DINOv2 generally demonstrate better performance for 3D properties than other networks trained at large scale: OpenCLIP, DINOv1, and VQGAN. *Third*, different time steps of Stable Diffusion

features, as well as different transformer layers of DINO/CLIP/VQGAN, perform best for different 3D physical properties.

Why is an understanding of the networks' ability to predict 3D properties useful? There are four reasons: (1) It begins to answer the scientific question of the extent to which these networks implicitly model the 3D scene; (2) The features we determined that are able to predict 3D physical properties can be used for this task, e.g. to predict shadow-object associations or support relations. This could either be carried out directly by incorporating them in a prediction network, in the manner of [51]; or they can be used indirectly as a means to train a feed forward network to predict the properties [42, 43]; (3) By knowing what properties Stable Diffusion is not good at, we have a way to spot images generated by Stable Diffusion, as has been done by [32]; (4) It also reveals which properties the network could be trained further on to improve its 3D modelling, *e.g.,* via extra supervision for that property.

## 2   Related Work

### 2.1   Exploration of Pre-trained Models

Building on the success of large-scale vision networks, there has been significant interest from the community to understand what has been learned by these complex models. On discriminative models, for example, [26, 49] propose inverse reconstruction to directly visualize the acquired semantic information in various layers of a trained classification network; [12, 13, 57] demonstrate that scene classification networks have remarkable localization ability despite being trained on only image-level labels; and [10, 34, 37] use the gradients of any target concept, flowing into the final convolutional layer to produce a saliency map highlighting important regions in the image for predicting the concept. In the more recent literature, [7] explores what has been learned in the powerful transformer model by visualizing the attention map.

On generative models, researchers have mainly investigated what has been learned in GANs, for example, GAN dissection [3] presents an analytic framework to visualize and understand GANs at the unit-, object-, and scene-level; [44] analyse the latent style space of StyleGANs [20]. The most recent work [32] studies the 3D geometric relations in generated images, such as vanishing points and shadows, and notes that the errors made can be used to discriminate real from generated images.

There is concurrent work [1] exploring the capability of predicting depth, surface normal and geometric correspondence for visual foundation models. To probe the depth and surface normal capability, a prediction network is trained using frozen multi-layer features from the foundation models. The capability for computing geometric correspondence between images is investigated using the dense spatial feature maps directly. In contrast to their work, we have studied a wider range of properties, covering both 3D geometric properties and 3D physical properties. Additionally, we have proposed a simple, general, yet efficient protocol for any property and any model, and have investigated the performance of different time steps and layers of models for different properties.

### 2.2   Exploitation of Pre-trained Models

Apart from understanding the representation in pre-trained models, there has been a recent trend for exploiting models trained at large scale, to tackle a series of downstream tasks. For example, leveraging generative models for data augmentation in recognition tasks [16, 19], utilising large vision models for semantic segmentation [2, 46], open-vocabulary segmentation [23], depth map estimation [21, 28, 35, 47, 48, 53, 56], correspondence estimation [17, 25, 27, 38, 54] and pose estimation [15, 55]. More recently, [4] searched for intrinsic offsets in a pre-trained StyleGAN for a range of downstream tasks, predicting normal maps, depth maps, segmentations, albedo maps, and shading.

### 2.3   3D Physical Scene Understanding

There have been works studying different 3D physical properties for scene understanding, including shadows [40, 41], material [39], occlusion [50], scene geometry [24], support relations [36] and depth [36]. However, these works focus on one or two physical properties, and most of them require training a model for the property in a supervised manner. In contrast, we use a single model to predict multiple properties, and do not train the features.

Table 1: **Overview of the datasets and training/evaluation statistics for the properties investigated.** For each property, we list the image dataset used, and the number of images for the train, val, and test set. 1000 images are used for testing if the original test set is larger than 1000 images. Regions are selected in each image, and pairs of regions are used for all the probe questions.

| Property: | | Same Plane | Perpendicular Plane | Material | Support Relation | Shadow | Occlusion | Depth |
|---|---|---|---|---|---|---|---|---|
| Dataset: | | ScanNetv2 | ScanNetv2 | DMS | NYUv2 | SOBA | Sep. COCO | NYUv2 |
| Images | # Train | 400 | 400 | 400 | 400 | 400 | 400 | 400 |
| | # Val | 100 | 100 | 100 | 100 | 100 | 100 | 100 |
| | # Test | 1000 | 1000 | 1000 | 654 | 160 | 983 | 654 |
| Regions | # Train | 7600 | 4493 | 4997 | 8943 | 3576 | 6799 | 8829 |
| | # Val | 1844 | 1112 | 1180 | 1968 | 976 | 1677 | 2075 |
| | # Test | 17159 | 10102 | 11364 | 13968 | 1176 | 16993 | 13992 |
| Pairs | # Train | 14360 | 17530 | 18520 | 13992 | 7152 | 19238 | 15322 |
| | # Val | 3498 | 4232 | 4284 | 2874 | 1952 | 4724 | 3786 |
| | # Test | 32654 | 38640 | 41824 | 21768 | 2352 | 44266 | 24880 |

# 3 Method – Properties, Datasets, and Classifiers

Our goal is to examine the ability of large-scale vision models to understand different physical and geometrical properties of the 3D scene, including: scene geometry, material, support relations, shadows, occlusion and depth. Specifically, we conduct linear probing of the features from different layers and time steps of the Stable Diffusion model, and different transformer layers for other models including OpenCLIP, DINOv1, DINOv2 and VQGAN. First, we set up the questions for each property (Section 3.1); and then select real image datasets with ground truth annotations for each property (Section 3.2). We describe how a classifier is trained to answer the questions, and the grid search for the optimal time step and layer to extract a feature for predicting the property in Section 3.3.

## 3.1 Properties and Questions

Here, we study the large vision model's ability to predict different *properties* of the 3D scene; the properties cover the 3D structure and material, the lighting, and the viewpoint. For each property, we propose *questions* that classify the relationship between a pair of *Regions*, *A* and *B*, in the same image, based on the features extracted from the large vision model. The properties and questions are:

- *Same Plane*: 'Are Region *A* and Region *B* on the same plane?'

- *Perpendicular Plane*: 'Are Region *A* and Region *B* on perpendicular planes?'

- *Material*: 'Are Region *A* and Region *B* made of the same material?'

- *Support Relation*: 'Is Region *A* (object *A*) supported by Region *B* (object *B*)?'

- *Shadow*: 'Are Region *A* and Region *B* in an object-shadow relationship?'

- *Occlusion*: 'Are Region *A* and Region *B* part of the same object but, separated by occlusion?'

- *Depth*: 'Does Region *A* have a greater average depth than Region *B*?'

We choose these properties as they exemplify important aspects of the 3D physical scene: the *Same Plane* and *Perpendicular Plane* questions probe the 3D scene geometry; the *Material* question probes what the surface is made of, *e.g.,* metal, wood, glass, or fabric, rather than its shape; the *Support Relation* question probes the physics of the forces in the 3D scene; the *Shadow* question probes the lighting of the scene; the *Occlusion* and *Depth* questions depend on the viewpoint, and probe the disentanglement of the 3D scene from its viewpoint.

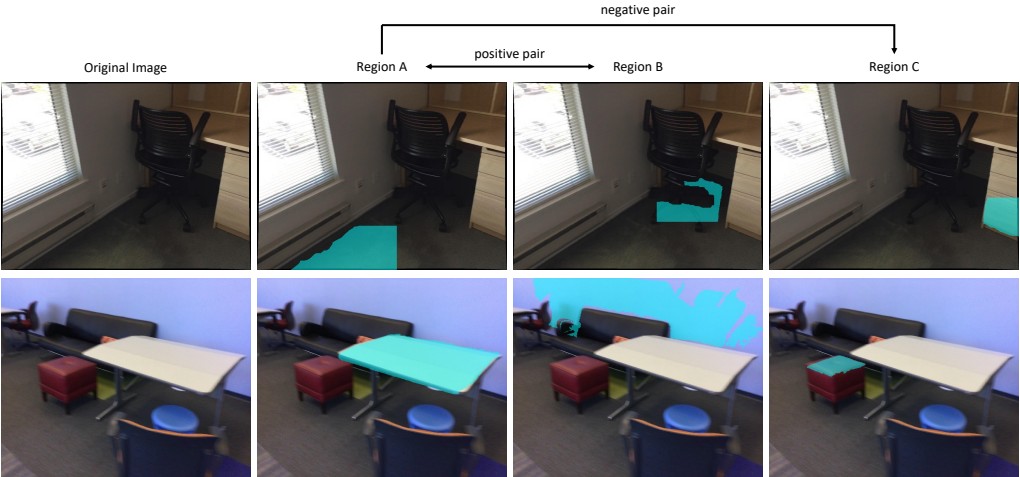

Figure 2: **Example images for probing *scene geometry*.** The first row shows a sample annotation for the *same plane*, and the second row is a sample annotation for *perpendicular plane*. Here, and in the following figures, (*A*, *B*) are a positive pair, while (*A*, *C*) are negative. The images are from the ScanNetv2 dataset [8] with annotations for planes from [24]. In the first row, the first piece of floor (*A*) is on the same plane as the second piece of floor (*B*), but is not on the same plane as the surface of the drawers (*C*). In the second row, the table top (*A*) is perpendicular to the wall (*B*), but is not perpendicular to the stool top (*C*).

## 3.2 Datasets

To study the different properties, we adopt various off-the-shelf real image datasets with annotations for the different properties, where the dataset used depends on the property. We repurpose each dataset to support probe questions of the form: $\mathcal{D} = \{(R_A, R_B, y)_1, \ldots, (R_A, R_B, y)_n\}$, where $R_A$, $R_B$ denote a pair of regions, and $y$ is the binary label indicating the answer to the considered question of the probed property. For each property, we create a train/val/test split from those of the original datasets, if all three splits are available. While for dataset with only train/test splits available, we divide the original train split into our train/val splits. Table 1 summarises the datasets used and the statistics of the splits used. We discuss each property and dataset in more detail next.

**Same Plane.** We use the ScanNetv2 dataset [8] with annotations of planes from [24]. Regions are obtained via splitting plane masks into several regions. A pair of regions are *positive* if they are on the same plane, and *negative* if they are on different planes. The first row of Figure 2 is an example.

**Perpendicular Plane.** We use the ScanNetv2 dataset [8]. We use the annotations from [24] which provide segmentation masks as well as plane parameters for planes in the image, so that we can obtain the normal of planes to judge whether they are perpendicular or not. A pair of regions are *positive* if they are on perpendicular planes, and *negative* if they are not on perpendicular planes. The second row of Figure 2 is an example.

**Material.** We adopt the recent DMS dataset [39] to study the material property, which provides dense annotations of material category for each pixel in the images. Therefore, we can get the mask of each material via grouping pixels with the same material label together. In total, there are 46 pre-defined material categories. Regions are obtained by splitting the mask of each material into different connected components, *i.e.,* they are simply groups with the same material labels, yet not connected. A pair of regions are *positive* if they are of the same material category, and *negative* if they are of different material categories. First row of Figure 3 is an example.

**Support Relation.** We use the NYUv2 dataset [36] to probe the support relation. Segmentation annotations for different regions (objects or surfaces) are provided, as well as their support relations. Support relation here means an object is physically supported by another object, *i.e.,* the second object will undertake the force to enable the first object to stay at its position. Regions are directly obtained via the segmentation annotations. A pair of regions are *positive* if the first region is supported by the

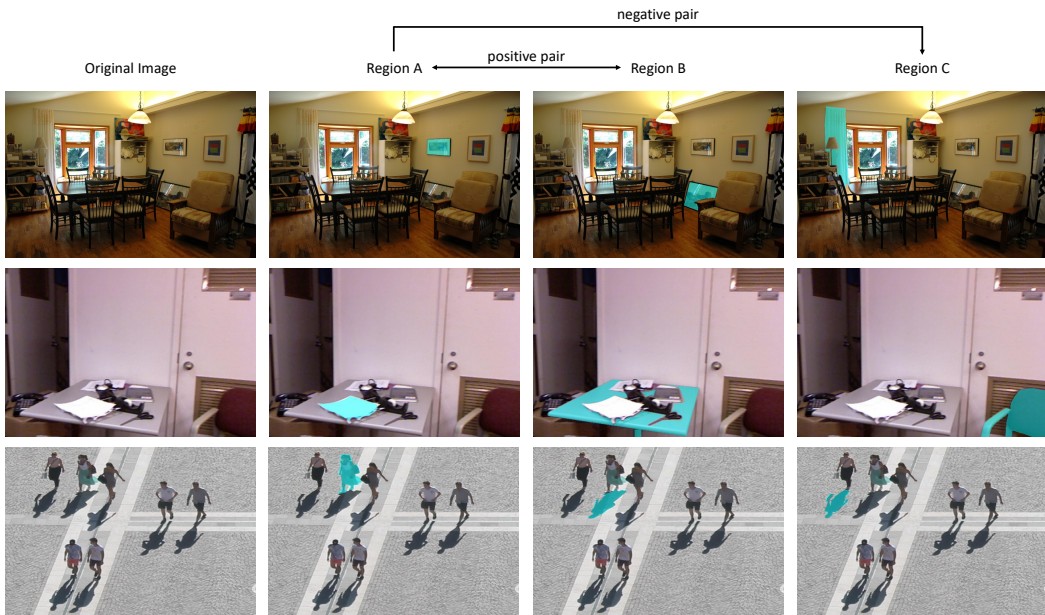

Figure 3: **Example images for probing *material, support relation and shadow*.** The first row is for *material*, the second row for *support relation*, and the third row for *shadow*. First row: the material images are from the DMS dataset [39]. The paintings are both covered with glass (*A* and *B*) whereas the curtain (*C*) is made of fabric. Second row: the support relation images are from the NYUv2 dataset [36]. The paper (*A*) is supported by the table (*B*), but it is not supported by the chair (*C*). Third row: the shadow images are from the SOBA dataset [41]. The person (*A*) has the shadow (*B*), not the shadow (*C*).

second region, and *negative* if the first region is not supported by the second region. Second row of Figure 3 is an example.

**Shadow.** We use the SOBA dataset [40, 41] to study the shadows which depend on the lighting of the scene. Segmentation masks for each object and shadow, as well as their associations are provided in the dataset annotations. Regions are directly obtained from the annotated object and shadow masks. In a region pair, there is one object mask and one shadow mask. A pair of regions are *positive* if the shadow mask is the shadow of the object, and *negative* if the shadow mask is the shadow of another object. Third row of Figure 3 is an example.

**Occlusion.** We use the Seperated COCO dataset [50] to study the occlusion (object seperation) problem. Regions are different connected components of objects (and the object mask if it is not separated), *i.e.,* groups of connected pixels belonging to the same object. A pair of regions are *positive* if they are different components of the same object separated due to occlusion, and *negative* if they are not from the same object. First row of Figure 4 is an example.

**Depth.** We use the NYUv2 dataset [36], which provides mask annotations for different objects and regions, together with depth for each pixel. A pair of regions are *positive* if the first region has a greater average depth than the second region, and *negative* if the first region has a less average depth than the second region. The average depth of a region is calculated via the average of depth value of each pixel the region contains. Second row of Figure 4 is an example.

### 3.3   Property Probing

Take Stable Diffusion as an example, we aim to determine which features best represent different properties. To obtain features from an off-the-shelf Stable Diffusion network, we follow the approach of [38] used for DIFT, where noise is added to the input image in the latent space, and features are extracted from different layers and time steps of the model. While probing the properties, linear classifiers are used to infer the relationships between *regions*. The region representation is obtained by a simple average pooling of the diffusion features over the annotated region or object.

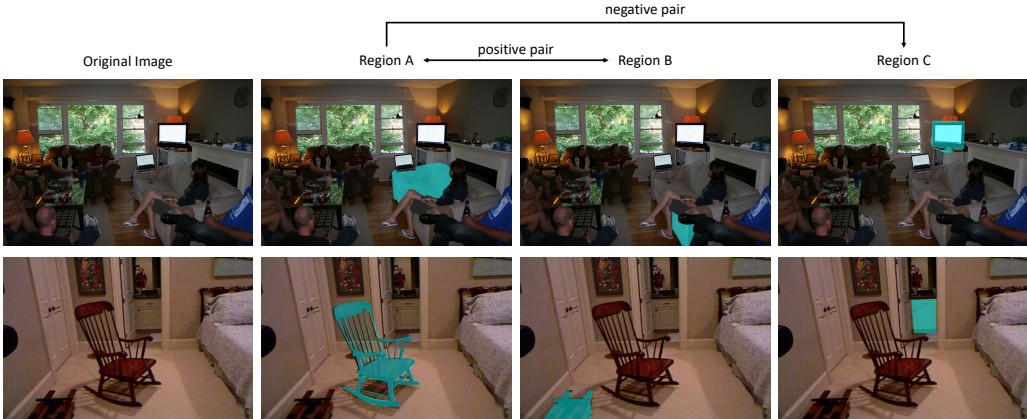

Figure 4: **Example images for probing *viewpoint-dependent properties (occlusion & depth)*.** The first row is for *occlusion* and the second row is for *depth*. First row: the occlusion images are from the Separated COCO dataset [50]. The sofa (*A*) and the sofa (*B*) are part of the same object, whilst the monitor (*C*) is not part of the sofa. Second row: the depth images are from the NYUv2 dataset [36]. The chair (*A*) is farther away than the object on the floor (*B*), but it is closer than the cupboard (*C*).

**Extracting Stable Diffusion Features.** We add noise $\epsilon \sim \mathcal{N}(0, \mathbf{I})$ of time step $t \in [0, T]$ to the input image $x_0$'s latent representation $z_0$ encoded by the VAE encoder:

$$z_t = \sqrt{\alpha_t} z_0 + (\sqrt{1 - \alpha_t})\epsilon \tag{1}$$

and then extract features from the immediate layers of a pre-trained diffusion model, $f_\theta(\cdot)$ after feeding $z_t$ and $t$ in $f_\theta$ ($f_\theta$ is a U-Net consisting of 4 downsampling layers and 4 upsampling layers):

$$F_{t,l} = f_{\theta_l}(z_t, t) \tag{2}$$

where $f_{\theta_l}$ is the $l$-th U-Net layer. In this way, we can get the representation of an image $F_{t,l}$ at time step $t$ and $l$-th U-Net layer for the probe. We upsample the obtained representation to the size of the original image with bi-linear, then use the region mask to get a region-wise feature vector, by averaging the feature vectors of each pixel it contains, *i.e.,* average pooling.

$$v_{k,t,l} = \text{avgpool}(R_k \odot \text{upsample}(F_{t,l})) \tag{3}$$

where $v_{k,t,l}$ is the feature vector of the $k$-th region $R_k$. $\odot$ here is a per-pixel product of the region mask and the feature. For other models, including CLIP, DINOv1, DINOv2 and VQGAN, we feed the image into the ViT/Transformer and extract features from different layers. Then use average pooling as in Equation 3 to obtain the feature for each region.

**Linear Probing.** After extracting features from large-scale vision models, we use a linear classifier (a linear SVM) to examine how well these features can be used to answer questions to each of the properties. Specifically, the input of the classifier is the difference or absolute difference between the feature vectors of Region *A* and Region *B*, *i.e.,* $v_A - v_B$ or $|v_A - v_B|$, and the output is a Yes/No answer to the question. Denoting the answer to the question as $Q$, then since the questions about *Same Plane*, *Perpendicular Plane*, *Material*, *Shadow* and *Occlusion* are symmetric relations, $Q(v_A, v_B) = Q(v_B, v_A)$. However, the questions about *Support Relation* and *Depth* are not symmetric. Thus, we use $|v_A - v_B|$ (a symmetric function) as input for the first group of questions, and $v_A - v_B$ (non-symmetric) for the rest of questions. We train the linear classifier on the train set via the positive/negative samples of region pairs for each property; do a grid search on the validation set to find (i) the optimal time step $t$ (only for Stable Diffusion), (ii) the U-Net layer $l$ for Stable Diffusion and the Transformer layer $l$ for other models, and (iii) the SVM regularization parameter $C$; and evaluate the performance on the test set. The grid search is only feasible because our proposed protocol is lightweight, and can assess the effectiveness of features for different downstream tasks with minimal resource demands.

# 4 Experiments

In this section, we first give details of the grid search method in Section 4.1. We then give the linear probing grid search results on features from Stable Diffusion in Section 4.2 and from other networks trained at scale in Section 4.3. Finally, we compare all models on the test set in Section 4.4.

## 4.1 Implementation Details and Evaluation Metric

**Implementation Details.** For each property, we sample the same number of positive / negative pairs, to maintain a balanced evaluation set. In terms of the linear SVM, we tune the penalty parameter $C$ on the val split to find the best $C$ for each property. Therefore, we are grid searching 3 parameters on the val set, namely, Timestep $t$ ranging from 1 to 1000 (only for Stable Diffusion), U-Net Layer $l$ covering the 4 downsampling and 4 upsampling layers for Stable Diffusion and Transformer Layer $l$ for other networks, and the SVM penalty parameter $C$ ranging over $0.001, 0.01, 0.1, 1, 10, 100, 1000$. The timestep is searched with a stride of 20 steps, since the difference in performance around the optimal value varies by less than 0.01 within 20 steps. In practice the $C$ parameter is always between 0.1 and 1, so we carry out a finer search over values between 0.1 and 1.0 in steps of 0.1. The linear SVM is solved using the *libsvm* library [6] with the SMO algorithm, to get the unique global optimal solution. Please refer to the appendix for more implementation details.

**Evaluation Metric.** All protocols are binary classification, therefore, we use ROC Area Under the Curve (AUC Score) to evaluate the performance of the linear classifier, as it is not sensitive to different decision thresholds.

## 4.2 Results for Stable Diffusion

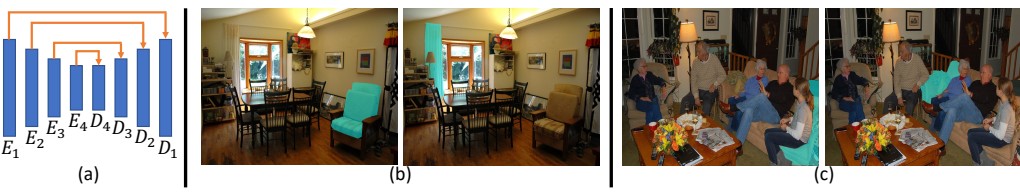

Figure 5: **(a) Nomenclature for the U-Net Layers.** We probe 4 downsampling encoder layers $E_1$-$E_4$ and 4 upsampling decoder layers $D_1$-$D_4$ of the Stable Diffusion U-Net. **(b) A prediction failure for *Material*.** In this example the model does not predict that the two regions are made of the same material (fabric). **(c) A prediction failure for *Occlusion*.** In this example the model does not predict that the two regions belong to the same object (the sofa).

Table 2: **SVM grid search results of Stable Diffusion features.** For each property, we train the linear SVM on the training set and grid search the best combination of time step, layer, and $C$ on the validation set. The ROC AUC score (%) is reported on the validation set using the selected combination.

| Property | Time Step | Layer | $C$ | Val AUC |
|---|---|---|---|---|
| Same Plane | 360 | $D_3$ | 0.4 | 97.3 |
| Perpendicular Plane | 160 | $D_3$ | 0.5 | 88.5 |
| Material | 20 | $D_3$ | 0.5 | 81.5 |
| Support Relation | 120 | $D_3$ | 1.0 | 92.6 |
| Shadow | 160 | $D_3$ | 0.8 | 95.4 |
| Occlusion | 180 | $D_3$ | 0.3 | 83.8 |
| Depth | 60 | $D_3$ | 0.9 | 99.2 |

The results for grid search are shown in Table 2. For Stable Diffusion U-Net Layer, $D_l$ means the $l$-th layer of the U-Net decoder, *i.e.,* upsampling layer, from outside to inside (right to left), and we provide an illustration of the layers in Figure 5(a).

We can make observations: First, the best time step for different properties varies, but the optimal time step is usually before 400. This is expected as a large time step adds too much noise, so not

much information is contained about the image. Second, in terms of the layer, the best U-Net layer is always $D_3$ in the decoder rather than the encoder. The optimal layer is in the middle, as $D_1$ is too close to the noise space and $D_4$ has just started decoding. Further explorations using Stable Diffusion features for downstream tasks could thus start from the U-Net decoder layers, especially $D_3$. Third, in terms of the performance on the test set, we find that Stable Diffusion can understand very well about scene geometry, support relations, shadows, and depth, but it is less performant at predicting material and occlusion. Examples of its failure are shown in Figure 5 (b) (c). As noted in [50] and [22], grouping all separated parts of an object due to occlusion remains challenging even for state-of-the-art detection and segmentation models. The appendix gives grid search results at all time steps and layers.

### 4.3 Results for CLIP/DINO/VQGAN Features

Table 3: **SVM grid search results of CLIP/DINO/VQGAN features.** We train the linear SVM on the training set, and grid search the best combination of ViT/Transformer layer and $C$ on the validation set. The OpenCLIP and VQGAN models we use have 48 transformer layers, DINOv1 has 12 layers and DINOv2 has 40 layers. The $i$-th layer means the $i$-th transformer layer from the input side.

| | Same Plane | | | | Perpendicular Plane | | | |
|---|---|---|---|---|---|---|---|---|
| | OpenCLIP | DINOv1 | DINOv2 | VQGAN | OpenCLIP | DINOv1 | DINOv2 | VQGAN |
| Optimal Layer | 27 | 8 | 24 | 12 | 27 | 9 | 22 | 12 |
| Optimal C | 0.7 | 0.7 | 0.8 | 1.0 | 1.0 | 0.2 | 0.6 | 0.6 |
| Val AUC | 94.5 | 93.2 | 96.0 | 82.6 | 72.9 | 70.9 | 84.9 | 62.8 |

| | Material | | | | Support Relation | | | |
|---|---|---|---|---|---|---|---|---|
| | OpenCLIP | DINOv1 | DINOv2 | VQGAN | OpenCLIP | DINOv1 | DINOv2 | VQGAN |
| Optimal Layer | 30 | 8 | 23 | 11 | 32 | 9 | 40 | 14 |
| Optimal C | 0.3 | 0.2 | 0.6 | 0.3 | 0.3 | 0.3 | 0.6 | 0.4 |
| Val AUC | 77.5 | 77.4 | 81.3 | 65.8 | 92.0 | 91.5 | 93.6 | 85.4 |

| | Shadow | | | | Occlusion | | | |
|---|---|---|---|---|---|---|---|---|
| | OpenCLIP | DINOv1 | DINOv2 | VQGAN | OpenCLIP | DINOv1 | DINOv2 | VQGAN |
| Optimal Layer | 28 | 2 | 29 | 8 | 31 | 3 | 29 | 2 |
| Optimal C | 1.0 | 0.8 | 1.0 | 1.0 | 0.2 | 0.2 | 0.3 | 1.0 |
| Val AUC | 94.6 | 92.4 | 96.6 | 88.7 | 80.6 | 77.0 | 84.4 | 77.4 |

| | Depth | | | |
|---|---|---|---|---|
| | OpenCLIP | DINOv1 | DINOv2 | VQGAN |
| Optimal Layer | 32 | 7 | 30 | 45 |
| Optimal C | 0.1 | 0.4 | 1.0 | 0.5 |
| Val AUC | 99.2 | 97.4 | 99.6 | 93.7 |

In this section we show grid search results for OpenCLIP [18, 30] pre-trained on LAION dataset [33], DINOv1 [5] pre-trained on ImageNet dataset [9], DINOv2 [27] pre-trained on LVD-142M dataset [27], and VQGAN [11] pre-trained on ImageNet dataset [9]. We use the best pre-trained checkpoints available on their official GitHub – ViT-B for DINOv1, ViT-G for OpenCLIP and DINOv2, and the 48-layer transformer checkpoint for VQGAN. Similar to Stable Diffusion, for each of these models, we conduct a grid search on the validation set in terms of the ViT/Transformer layer and $C$ for SVM, and use the best combination of parameters for evaluation on the test set.

Grid search results are reported in Table 3. It can be observed that different layers of different models are good at different properties.

### 4.4 Comparison of Different Features Trained at Scale

Table 4: **Comparison of different features trained at scale.** For each property, we use the best time step, layer and $C$ found in the grid search for Stable Diffusion, and the best layer and $C$ found in the grid search for other features. The performance is the ROC AUC on the test set, and 'Random' means a random classifier.

| Property | Random | OpenCLIP | DINOv1 | DINOv2 | VQGAN | Stable Diffusion |
|---|---|---|---|---|---|---|
| Same Plane | 50 | 92.3 | 91.4 | 94.5 | 81.3 | **96.3** |
| Perpendicular Plane | 50 | 71.8 | 71.3 | 82.1 | 62.0 | **86.0** |
| Material | 50 | 78.7 | 78.8 | 83.5 | 65.5 | **83.6** |
| Support Relation | 50 | 90.6 | 90.8 | **92.8** | 84.1 | 92.1 |
| Shadow | 50 | 94.9 | 92.2 | **95.8** | 89.0 | 95.4 |
| Occlusion | 50 | 81.2 | 79.9 | 84.4 | 78.4 | **84.8** |
| Depth | 50 | 99.2 | 97.1 | **99.7** | 91.8 | 99.6 |

We compare the state-of-the-art pre-trained large-scale vison models' representations on various downstream tasks in Table 4. It can be observed that the Stable Diffusion and DINOv2 representations outperform OpenCLIP, DINOv1 and VQGAN for all properties, indicating the potential of utilizing Stable Diffusion and DINOv2 representations for different downstream tasks with the optimal time steps and layers found in Section 4.2 and Section 4.3.

## 5 Discussion and Future Work

In this paper, we have developed a general and lightweight protocol to efficiently examine whether models pre-trained on large scale image datasets, like CLIP, DINO, VQGAN and Stable Diffusion, have explicit feature representations for different properties of the 3D physical scene.

It is interesting to find that different time steps of Stable Diffusion and different layers of DINOv2 representations can handle several different physical properties at a state-of-the-art performance, indicating the potential of utilising the Stable Diffusion and DINOv2 models for different downstream tasks.

However, for some properties such as material and occlusion, the networks are not distilling the information in a manner that can be used by a linear probe. This could indicate that these properties are not modelled well by the network or that more than a linear probe is required to tease them out. We show examples of the prediction failures for these properties in Figure 5. In the appendix, we show that such prediction failures also occur in generated (i.e. synthetic) images. It is worth noting that occlusion is a challenge even for the powerful Segment Anything Model (SAM) [22], where the model 'hallucinates small disconnected components at times'.

In the appendix, we provide preliminary results of using the probed Stable Diffusion feature for downstream tasks (Section E). We also provide examples of another use case of spotting Stable Diffusion generated images based on the properties that the model is not good at (Section B).

This paper has given some insight into answering the question: 'To what extent do large vision models understand the 3D scene' for real images. Of course, there are more properties that could be investigated in the manner proposed here. For example, contact relations [14] and object orientation [45], as well as the more nuanced non-symmetric formulations of the questions. Please refer to arxiv version of the paper [52] for future updates.

**Broader Impact.** This paper studies the learned representations of large vision models, and can help people better understand what has been learnt by these vision foundation models. We do not think there is any negative societal impact from this investigation.

**Acknowledgements.** This research is supported by EPSRC Programme Grant VisualAI EP/T028572/1, a Royal Society Research Professorship RP\R1\191132, an AWS credit funding, a China Oxford Scholarship and ERC-CoG UNION 101001212. We thank Yash Bhalgat, Minghao Chen, Subhabrata Choudhury, Kai Han, Tengda Han, Jaesung Huh, Vladimir Iashin, Tomas Jakab, Gyungin Shin, Ashish Thandavan, Vadim Tschernezki, Jianyuan Wang and Yan Xia from the Visual Geometry Group for their help and support for the project.

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

# Appendix

## A   More Implementation Details

**Extracting Stable Diffusion Features.**   Following DIFT [38], when we extract Stable Diffusion features, we add a different random noise 8 times and then take the average of the generated features. The process can be completed in one forward pass as we are using a batch of 8. We use an empty prompt '' as the text prompt.

**Train/Val Partition.**   For the partition of train/val split, we select the train & val images from different scenes for the NYUv2 [36] and ScanNetv2 [8] dataset.

**Sampling of Images.**   For the train/val/test splits, if the number of images used is less than the original number of images in the datasets, we randomly sample our train/val/test images from the original datasets.

**Sampling of Positive/Negative Pairs.**   For each image, if the number of possible negative pairs is larger than the number of possible positive pairs, we randomly sample from the negative pairs to obtain an equal number of negative and positive pairs, and vice versa. In this way, we keep a balanced sampling of positive and negative pairs for the binary linear classifier. As can be observed in Table 1, the number of train/val pairs for different properties are different, although we keep the same number of train/val images for different properties. This is because for different properties the availability of positive/negative pairs are different. For *depth*, we select a pair only if the average depth of one region is 1.2 times greater than the other because it is even challenging for humans to judge the depth order of two regions below this threshold. For *perpendicular plane*, taking the potential annotation errors into account, we select a pair as perpendicular if the angle between their normal vectors is greater than 85°and smaller than 95°, and select a pair as not perpendicular if the angle between their normal vectors is smaller than 60°or greater than 120°.

**Region Filtering.**   When selecting the regions, we filter out the small regions, *e.g.,* regions smaller than 100 pixels, because regions that are too small are challenging even for humans to annotate.

**Image Filtering.**   As there are some noisy annotations in the [24] dataset, we manually filter the images whose annotations are inaccurate.

**Linear SVM.**   The feature vectors are L2-normalised before inputting into the linear SVM. The binary decision of the SVM is given by $sign(w^T v + b)$, where $v$ is the input vector to SVM:

$$v = |v_A - v_B| \tag{4}$$

for the *Same Plane*, *Perpendicular Plane*, *Material*, *Shadow* and *Occlusion* questions, and

$$v = v_A - v_B \tag{5}$$

for the *Support Relation* and *Depth* questions.

**Extension of Separated COCO.**   To study the occlusion problem, we utilise the Separated COCO dataset [50]. The original dataset only collects separated objects due to occlusion in the COCO 2017 val split, we further extend it to the COCO 2017 train split for more data using the same method as in [50]. As the original COCO val split is too small (only 5k images v.s. 118k images in the COCO train split), we get our partition of train/val/test splits by dividing the generated dataset ourselves.

**Computing Resources.**   We run experiments on CPU with 20G memory for SVM training/testing.

## B More Stable Diffusion Generated Images

Here we give more examples of Stable Diffusion generated images as mentioned in the caption of Figure 1. We show examples for: **Scene Geometry** in Figure 6; **Material**, **Support Relations**, and **Shadows** in Figure 7; and **Occlusion** and **Depth** in Figure 8.

The observations match our findings on studying the Stable Diffusion features – Stable Diffusion 'knows' about a number of physical properties including scene geometry, material, support relations, shadows, occlusion and depth, but may fail in some cases in terms of material and occlusion.

As mentioned in Section 5, we can spot Stable Diffusion generated images via the properties it is not good at. Take the second row of Figure 7 and the first row of Figure 8 as examples – we can spot that the images are generated by Stable Diffusion because of the failure to generate a clear boundary between two different materials, and the failure to connect separated parts due to occlusion.

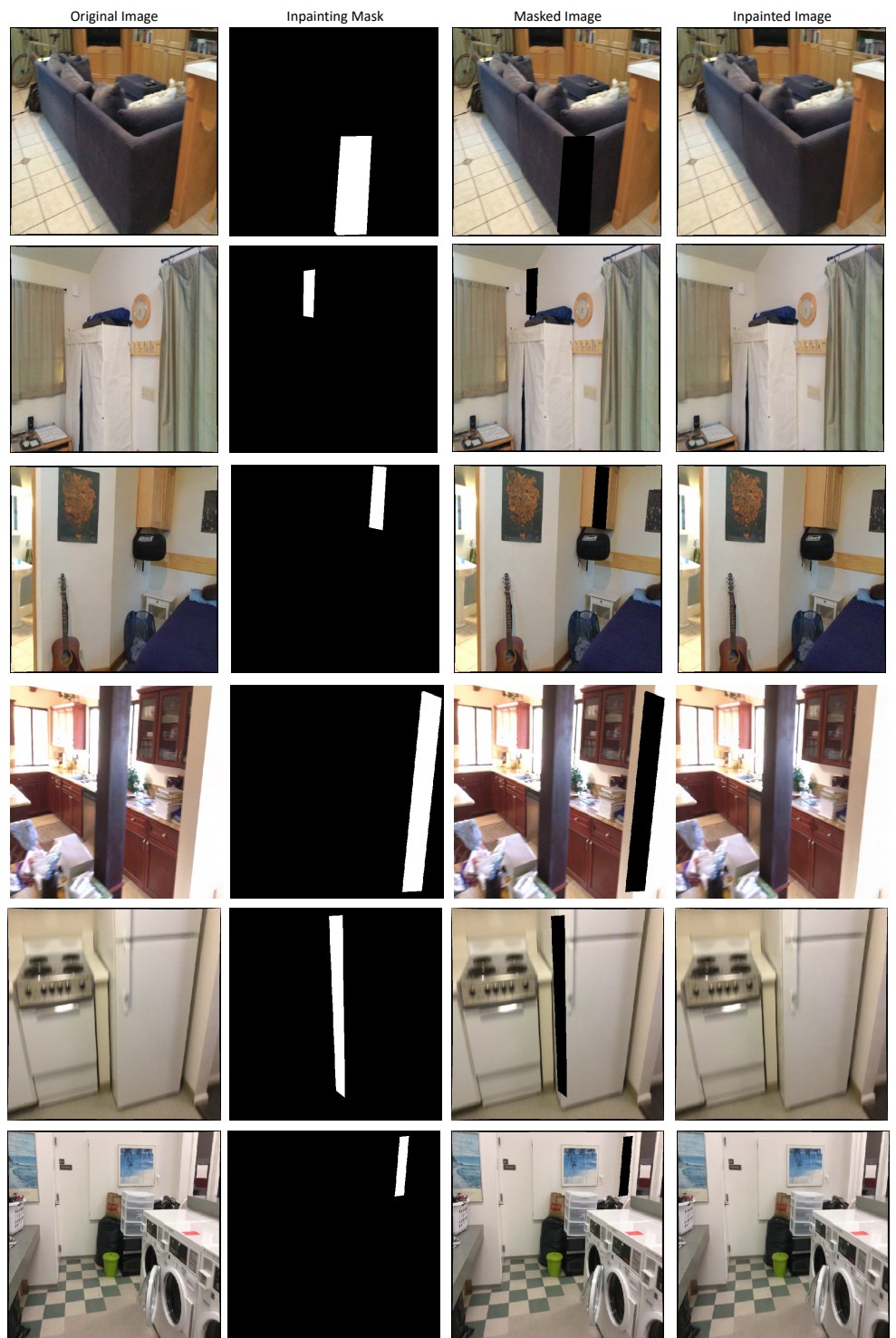

Figure 6: **Stable Diffusion generated images testing *scene geometry* prediction.** Here and for the following figures, the model is tasked with inpainting the masked region of the real images. Stable Diffusion 'knows' about *same plane* and *perpendicular plane* relations in the generation. When the intersection of two sofa planes (first row), two walls (second and sixth row), two cabinet planes (third row), two pillar planes (fourth row) or two fridge planes (fifth row) is masked out, Stable Diffusion is able to generate the two perpendicular planes at the corner based on the unmasked parts of the planes.

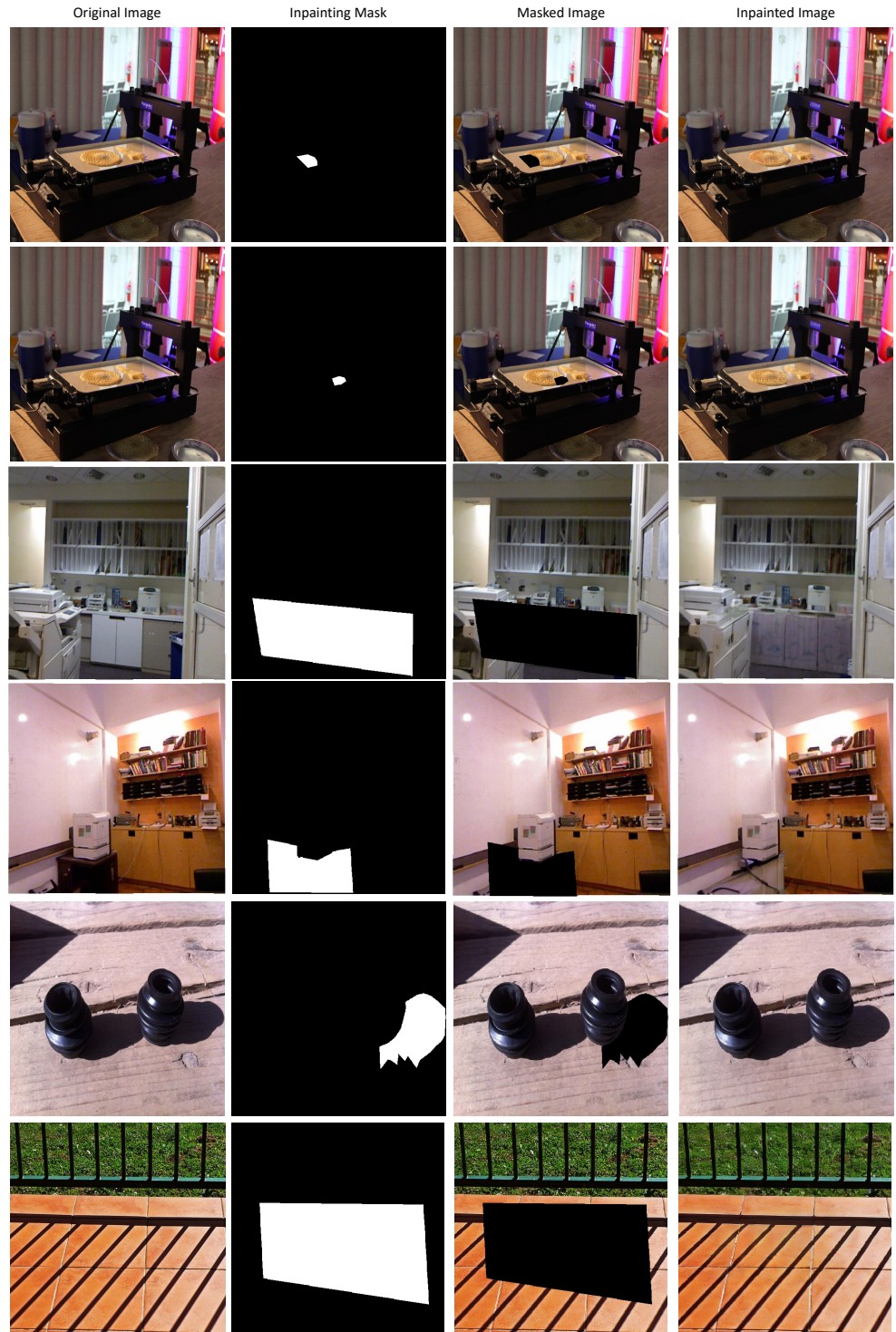

Figure 7: **Stable Diffusion generated images testing *material*, *support relation* and *shadow* prediction.** Stable Diffusion 'knows' about *support relations* and *shadows* in the generation, but may fail sometimes for *material*. Rows 1-2: Material; Rows 3-4: Support Relation; Rows 5-6: Shadow. In the first row, the model distinguishes the two different materials clearly and there is clear boundary between the generated pancake and plate; while in the second row, the model fails to distinguish the two different materials clearly, generating a mixed boundary. In the third row and fourth rows, the model does inpaint the supporting object for the stuff on the table and the machine. In the fifth and sixth rows, the model manages to inpaint the shadow correctly. Better to zoom in for more details.

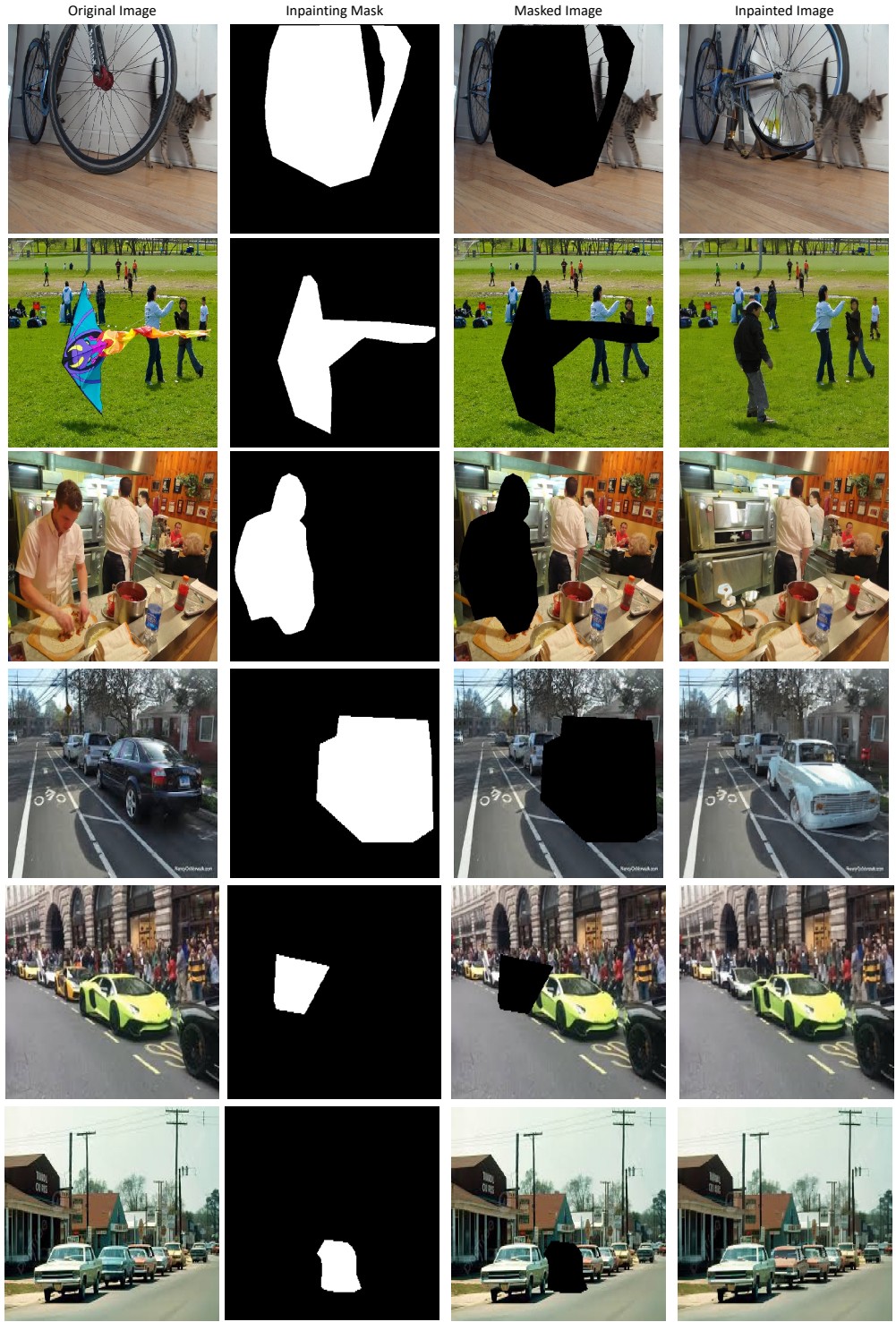

Figure 8: **Stable Diffusion generated images testing *occlusion* and *depth* prediction.** Stable Diffusion 'knows' about *depth* in the generation, but may fail sometimes for *occlusion*. Rows 1-3: Occlusion; Rows 4-6: Depth. In Row 1, the model fails to connect the tail with the cat body and generates a new tail for the cat, while in Row 2, the model successfully connects the separated people and generates their whole body, and in Row 3, the separated parts of oven are connected to generate the entire oven. In Rows 4-6, the model correctly generates a car of the proper size based on depth. The generated car is larger if it is closer, and smaller if it is farther away.

# C   Train/Val AUC Results of Stable Diffusion Features

Table 5 shows the train/val AUC of the SVM grid search results for Stable Diffusion features at the best combination of time step, layer and $C$ as in Table 4.2.

Table 5: **Train/Val AUC of SVM grid search for Stable Diffusion features.** For each property, the Train/Val AUC at the best combination of time step, layer and $C$ is reported.

| Property | Train AUC | Val AUC |
|---|---|---|
| Same Plane | 97.5 | 97.3 |
| Perpendicular Plane | 90.1 | 88.5 |
| Material | 87.6 | 81.5 |
| Support Relation | 93.7 | 92.6 |
| Shadow | 96.2 | 95.4 |
| Occlusion | 86.7 | 83.8 |
| Depth | 99.8 | 99.2 |

# D    AUC Curves for Stable Diffusion Features Grid Search

As mentioned in Section 4.2, we provide curves for AUC at different layers and time steps of probing Stale Diffusion features in Figure 9 - Figure 15.

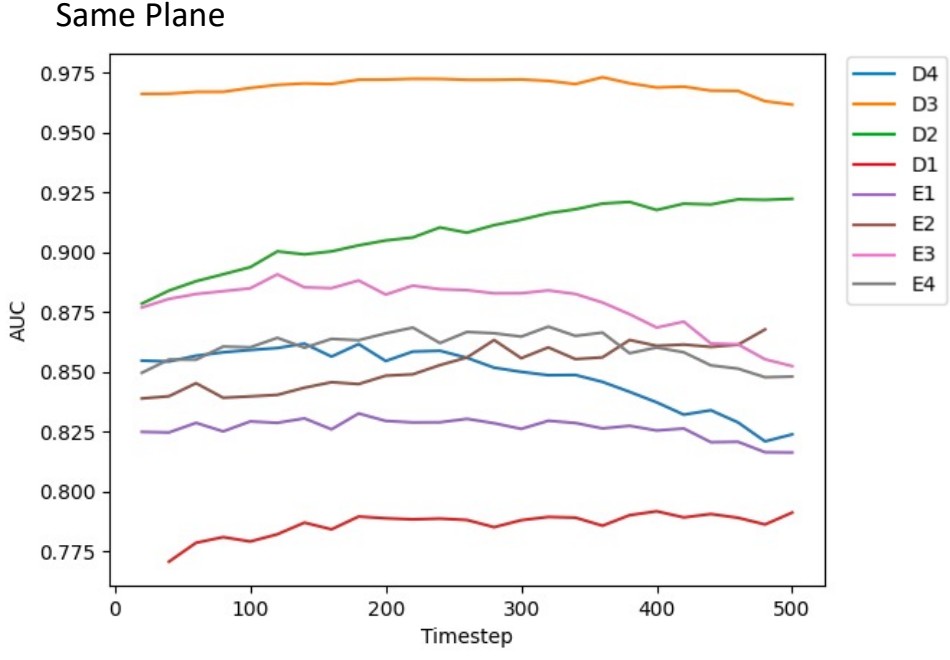

Figure 9: **Curves for AUC at different layers and time steps of probing Stable Diffusion for the** *same plane* **task.**

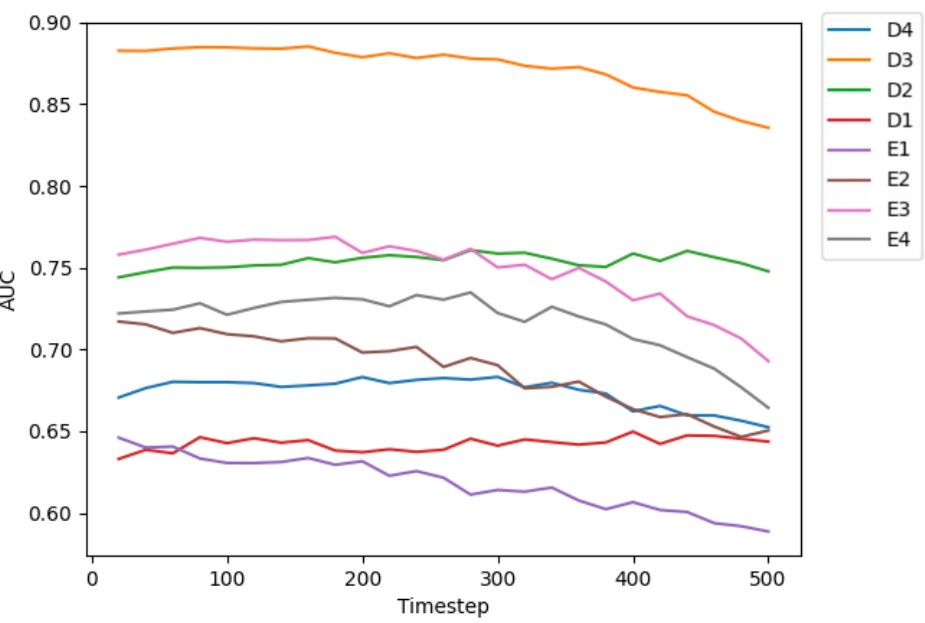

Figure 10: **Curves for AUC at different layers and time steps of probing Stable Diffusion for the** *perpendicular plane* **task.**

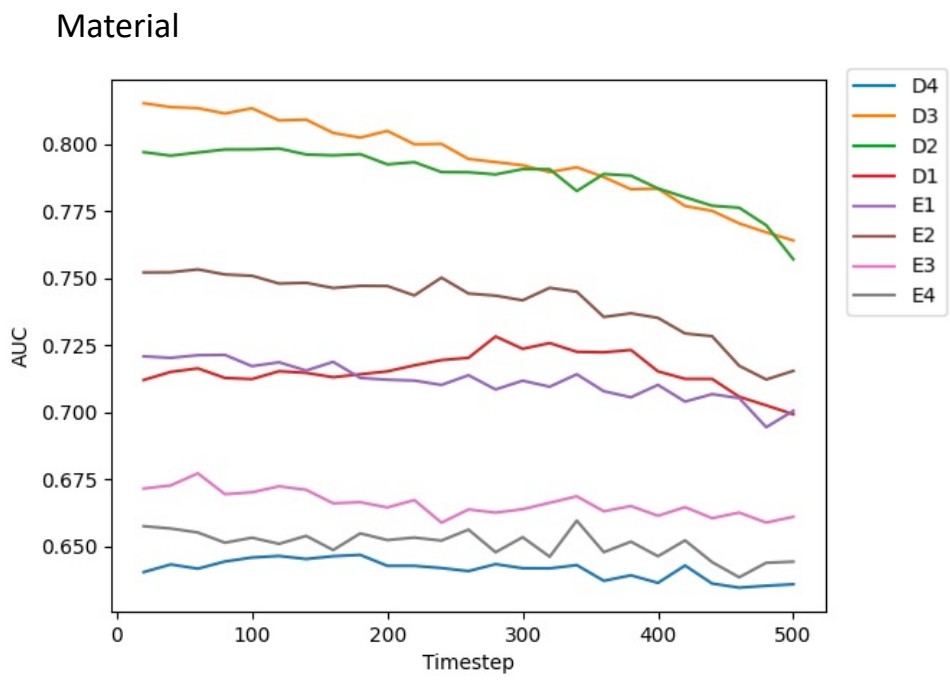

Figure 11: **Curves for AUC at different layers and time steps of probing Stable Diffusion for the** *material* **task.**

Support Relation

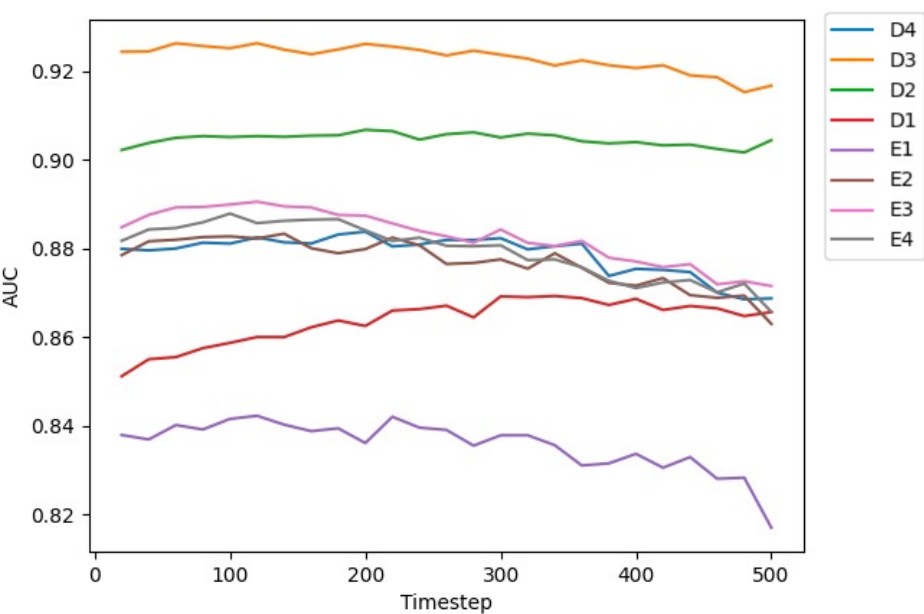

Figure 12: **Curves for AUC at different layers and time steps of probing Stable Diffusion for the** *support* **task.**

Shadow

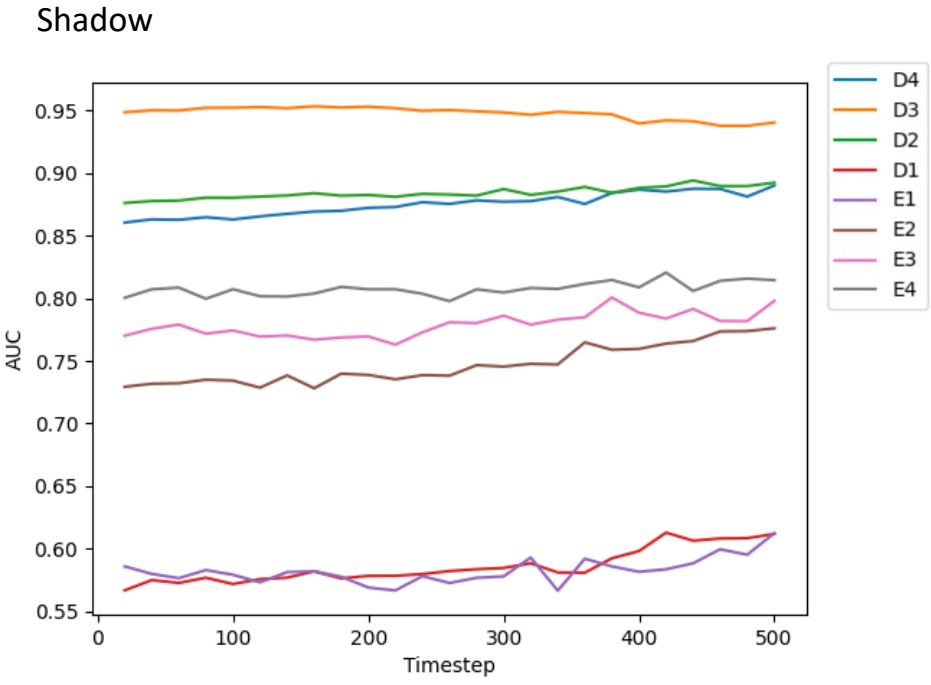

Figure 13: **Curves for AUC at different layers and time steps of probing Stable Diffusion for the** *shadow* **task.**

## Occlusion

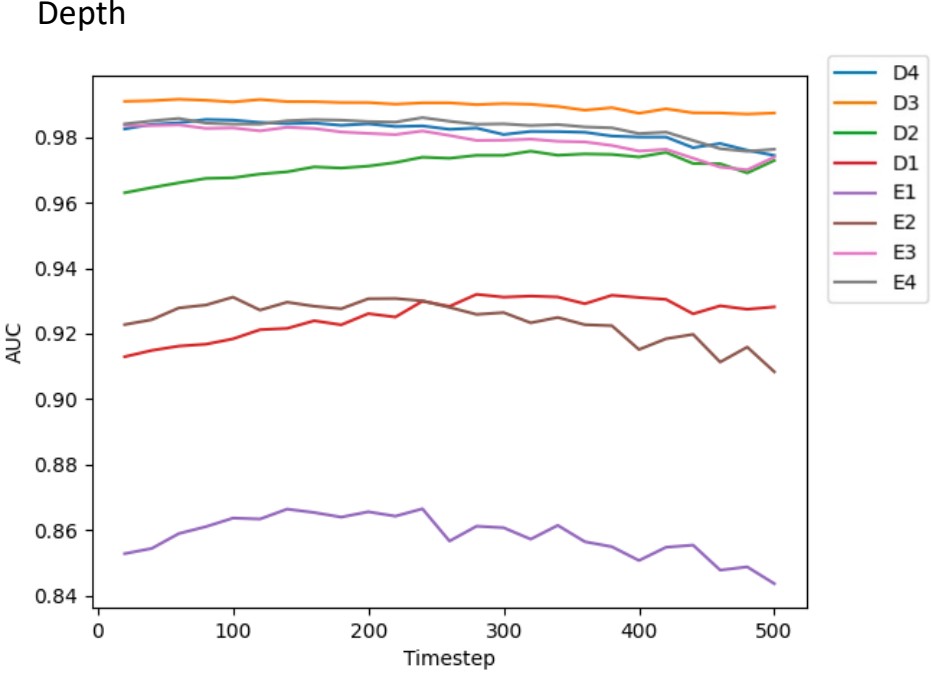

Figure 14: **Curves for AUC at different layers and time steps of probing Stable Diffusion for the** *occlusion* **task.**

## Depth

Figure 15: **Curves for AUC at different layers and time steps of probing Stable Diffusion for the** *depth* **task.**

# E Preliminary Results of Applying Probed Feature for Downstream Tasks

As mentioned in Section 5, we provide preliminary results of using the features selected by our probe for downstream tasks. We consider two tasks: surface normal estimation and depth estimation.

**Surface Normal Estimation.** For this, we use the specific time step and layer of the Stable Diffusion features selected by the 'perpendicular plane' probe. We directly inject the SD extracted features into the model iDisc [29] at the layer where the network encodes the image feature. When injecting Stable Diffusion feature and training for only 1000 iterations (whereas, [29] train 45000 iterations in total), the model performs better on most of the metrics compared with [29] reported results (see Table 6), which indicates that our feature probing strategy is relevant for corresponding 3D tasks.

Table 6: **Preliminary results of using the probed feature for the downstream task of normal estimation.** Here we show the results of injecting the selected Stable Diffusion feature into iDisc [29]. Please see text for more details.

| Model | Mean Angular Error $\downarrow$ | Angular RMSE $\downarrow$ | $\% < 11.25 \uparrow$ | $\% < 22.5 \uparrow$ | $\% < 30 \uparrow$ |
|-------|-------------------------|-----------------|-----------|----------|---------|
| iDisc | 14.6 | 22.8 | 63.8 | 79.8 | 85.6 |
| Ours | 13.8 | 18.8 | 58.1 | 80.9 | 88.7 |

**Depth Estimation.** We use the Stable Diffusion feature from the specific time step and layer (Table 2 Row 7) selected by the linear probe for depth, as the input to a simple convolutional network to predict the depth. The network is trained on the NYUv2 Depth training dataset with the SD features frozen. For comparison we train the same convolutional network using image features from a frozen ResNet-50 pre-trained on ImageNet. In Table 7 we can observe that the SD features have a substantially higher performance. This again illustrates the potential of the features selected by the probe for downstream applications.

Table 7: **Preliminary results of using the probed feature for downstream task of depth estimation.** Here we show the results of a comparison between ResNet and SD features on the NYUv2 Depth test dataset. Please see text for more details.

| Features | $\delta_1 \uparrow$ | $\delta_2 \uparrow$ | $\delta_3 \uparrow$ | RMSE $\downarrow$ |
|----------|------------|------------|------------|---------|
| ResNet-50 | 51.0 | 81.5 | 94.3 | 1.0065 |
| Ours(SD) | 80.8 | 97.2 | 99.6 | 0.5389 |

In summary, we have shown two examples where features from Stable Diffusion, selected by our linear probe, support downstream 3D tasks. Features could also be selected for other downstream tasks, including Instance Shadow Detection [41], Material Segmentation [39], and Support Relation Inference [36].

