# OpenReview forum: "A General Protocol to Probe Large Vision Models for 3D Physical Understanding"
_NeurIPS.cc/2024/Conference — NeurIPS 2024 poster_

### Official Review · Reviewer_Wno9 · 2024-07-10

**Soundness:** 3
**Presentation:** 4
**Contribution:** 4
**Rating:** 5
**Confidence:** 4

**Summary:**

This paper aims to evaluate how well large-scale vision models encode 3D properties of the scenes depicted in images. The paper proposes a general and lightweight protocol that involves training discriminative classifiers on the features of pre-trained models to assess their encoding of several physical properties of 3D scenes. These properties include scene geometry, materials, support relations, lighting, shadow, occlusion and depth.

This protocol is applied to several large vision models, such as CLIP, DINOv1, DINOv2, VQGAN, and Stable Diffusion. The findings indicate that features from Stable Diffusion and DINOv2 are particularly adept at discriminative learning for properties like scene geometry, support relations, shadows, and depth. However, they perform less well for occlusion and material properties. The results also show that different layers and time steps of these models are good at different properties, which could have potential applications for 3D physical understanding.

**Strengths:**

1. This paper proposes a protocol to assess the 3D awareness for current large-scale vision models. The protocol considers various physical properties and is lightweight.

2. The overall structure and writing are easy to follow.

3. The problem investigated in this paper is very interesting. It provides a new perspective to interpret the features learned by large-scale models, especially their 3D awareness. This is very valuable for future studies.

**Weaknesses:**

1. It seems unclear to me whether the proposed probes can really reflect the 3D understanding of large vision models, due to properties to be probed and the way to probe them. For example, the material, support relation and shadow properties, from my point of view, can be well identified with 2D clues (e.g. appearance and 2D spatial location). For occlusion and depth, using linear SVM to answer binary questions may not be enough to assess these properties.

2. There is a lack of baselines for the linear probing. It is unclear how a 3D-aware model trained explicitly with 3D data will respond to probe questions (upper bound). Also, it is better to show how the models with little 3D awareness will react to the probes (lower bound). Otherwise, it is less informative just comparing between large vision models, especially when their scores are close and pretty high as shown in table 4.

3. The paper “Probing the 3d awareness of visual foundation models” [1] shares a very similar goal with this paper. It’s suggested to add comparison and discussion with that paper.

4. It would be more comprehensive to evaluate more large vision models such as SAM [2] and MAE [3].

[1] Mohamed El Banani, Amit Raj, Kevis-Kokitsi Maninis, Abhishek Kar, Yuanzhen Li, Michael Rubinstein, Deqing Sun, Leonidas Guibas, Justin Johnson, and Varun Jampani. Probing the 3d awareness of visual foundation models. CVPR, 2024.

[2] Alexander Kirillov, Eric Mintun, Nikhila Ravi, Hanzi Mao, Chloe Rolland, Laura Gustafson, Tete Xiao, Spencer Whitehead, Alexander C Berg, Wan-Yen Lo, et al. Segment anything. ICCV, 2023.

[3] Kaiming He, Xinlei Chen, Saining Xie, Yanghao Li, Piotr Dollar, and Ross Girshick. Masked autoencoders are scalable vision learners. CVPR, 2022

**Questions:**

1. Have the large-scale models been trained on the dataset chosen in this paper? Does it have influence on the probe results?

2. In Table 4, instead of using a random classifier as the baseline, it is fairer to train a feature extractor from scratch with your probing training set.

3. Following the second point in Weakness, could you provide the upper bound and lower bound of the probe? That’s essential to justify that the probing tasks can actually reflect 3D awareness.

4. Currently, the input to the linear SVM is designed as the difference between the averaged pooled features from two patches. Will pooling and subtraction operations lead to information loss? Is there any better way to formulate the linear probing?

**Limitations:**

Yes.

---

> ### Author Rebuttal · Authors · 2024-08-06
>
> Thanks for your valuable comments. We have provided responses below:
>
> **W1: Whether the proposed probes can reflect the 3D understanding: The material, support relation and shadow properties can be well identified with 2D clues (appearance and 2D spatial location). For occlusion and depth, using linear SVM to answer the binary questions may not be enough for assessment.**
>
> We respectfully disagree that the material, support relation and shadow properties can be well identified with 2D clues. In the attached PDF, we show examples in our chosen datasets to demonstrate the challenge of identifying these properties from appearance and 2D spatial location. Additionally, the material, support relation and shadow datasets have also been used by others for 3D tasks and evaluation [1,2,3].
>
> For occlusion and depth, it is natural to formulate the questions in a binary way: the model can understand the separated instances if and only if it can figure out whether two regions belong to the same object or not; the model can understand depth if and only if it can figure out in a pair which region has a greater depth. In addition, the effectiveness of our probe for occlusion and depth can be further validated by: 1) As in Lines 66-67 and Figure 8, the observation of SD generated images confirms the conclusion of our linear probe; 2) As in Lines 62-65, the features we probed can be also used for downstream 3D applications, such as depth, as in the response to [Reviewer Lz9q W1].
>
>
> **W2 & Q3: Probe 3D-aware model trained explicitly with 3D data as upper bound; probe model with little 3D awareness as lower bound; scores are close and pretty high in Table 4**
>
> For an upper bound on the depth task, we evaluate the DepthAnythingv2 features, as the model has been explicitly trained with depth data and is the state-of-the-art depth prediction model. The result is 99.8 AUC on the test split, higher than any of the models we evaluated.
>
> It is difficult to do a lower bound since, as we have shown, models trained on image objectives (e.g. DINOv2) do develop a 3D awareness. Among them, VQGAN may be a lower bound as it is only focusing on simple 2D generation and trained on relatively simple dataset.
>
> The scores for different models in our paper are not 'close and pretty high in Table 4', e.g., for the material, perpendicular and occlusion tasks. For example, for the perpendicular task, the lowest performance is 62.0 while the highest performance is 86.0
>
>
> **W3: Comparison and discussion with the paper “Probing the 3d awareness of visual foundation models”**
>
> We have already discussed and compared with this concurrent paper in Lines 84-89 of the Related Work.
>
>
> **W4: Evaluate more large vision models such as SAM and MAE**
>
> Given the limited time in the rebuttal period, we have evaluated SAM for the depth and perpendicular tasks (i.e. we have used the linear probe to select features from the SAM ViT encoder). The results are 95.8 AUC for depth (lower than DINOv2/SD). We have not evaluated MAE as it is well known that MAE requires fine-tuning for downstream tasks [4,5], so is unlikely to be suitable.
>
>
> **Q1: Have the large-scale models been trained on the dataset chosen in this paper? Does it have influence on the probe results?**
>
> No. The large-scale models have not been trained on the datasets chosen in the paper. The datasets used for evaluation are listed in the second row of Table 1. To our knowledge, these do not overlap with LAION (for CLIP and Stable Diffusion), LVD-142M (for DINOv2), ImageNet (for DINOv1 and VQGAN).
>
>
> **Q2: Train a feature extractor from scratch on our probing training set as a baseline**
>
> We have conducted experiments on a model trained from scratch on our chosen shadow dataset SOBA. The model architecture is in Figure 4 of [6] and is trained with ground truth annotations of the associated shadow and instance masks. We extract features from the image encoder. The result on the test set is 73.8 AUC, which is lower than the models we have evaluated.
>
>
> **Q4: Will pooling and subtraction lead to information loss?**
>
> We must do pooling to unify the region features as vectors. If not, the dimension will be too big. For subtraction, we have conducted experiments using concatenation instead for the depth and support tasks for SD features, and it does not make much difference to the results: for depth 99.6 (concatenation) v.s. 99.6 (subtraction); for support 92.4 (concatenation) v.s. 92.1 (subtraction).
>
>
>
> [1] MaPa: Text-driven Photorealistic Material Painting for 3D Shapes. Zhang et al. SIGGRAPH 2024
>
> [2] Support surface prediction in indoor scenes. Guo et al. ICCV 2013
>
> [3] What You Can Reconstruct from a Shadow. Liu et al. CVPR 2023
>
> [4] ViTDet: Exploring Plain Vision Transformer Backbones for Object Detection. Li et al. ECCV 2022
>
> [5] Benchmarking Detection Transfer Learning with Vision Transformers. Li et al. arxiv 2021
>
> [6] Instance Shadow Detection with A Single-Stage Detector. Wang et al. TPAMI 2022

---

> > ### Comment · Reviewer_Wno9 · 2024-08-12
> >
> > Thanks for the rebuttal and I would maintain my positive rating.

---

### Official Review · Reviewer_4isv · 2024-07-10

**Soundness:** 3
**Presentation:** 3
**Contribution:** 4
**Rating:** 7
**Confidence:** 4

**Summary:**

This paper investigates several mainstream large vision models for their 3D physical understanding. The authors curated a binary classification benchmark covering a set of important 3D properties based on publicly available 2D image datasets and linearly probed different layers and different time steps of the vision models. Consequently, the authors find that DINOv2 and Stable Diffusion perform the best among the probed models, while some properties such as material and occlusion challenge them all. Consequently, this work provides a general linear probing protocol and dataset to evaluate a vision model's capability in 3D physical understanding.

**Strengths:**

1. Significance. The reviewer thinks this work studies a significant question. With the emergence of large vision models trained on 2D images, how and how well they could be applied to 3D-related tasks is important and of growing interest.
2. Originality. This work defines a set of properties and corresponding tasks for evaluating 3D physical understanding. This is a novel benchmark for evaluating vision models.
3. Clarity. The writing is clear and easy to follow. Section 3 presents clear demonstrations of the datasets and task definitions.
4. Quality. Through and organized experiments are conducted. Grid search is applied to timesteps and layers.

**Weaknesses:**

1. Lack of in-depth analysis and understanding. While the paper spends a big part describing the methods and experiments, an analysis of the performance differences seems to be glossed over. As many of the popular vision models are benchmarked, what are the authors' speculations and thoughts on the cause of the varied performance? Does the difference in data and training objectives play a part? The reviewer believes providing further analysis and understanding of the models will make this paper appreciated by more audiences.

**Questions:**

- For curating the dataset, how are the regions selected? For those obtained from annotation masks, are all the regions kept in the final dataset? Is there any human intervention and selection?
- Does the size of regions affect the task difficulty?
- Does patch size affect the performance? What are the patch sizes of the evaluated models?

**Limitations:**

The authors have discussed some limitations and broader impacts in Section 5.

---

> ### Author Rebuttal · Authors · 2024-08-06
>
> Thanks for your valuable comments. We have provided responses below:
>
> **W1: In-depth analysis and understanding: What’s the speculations and thoughts on the cause of varied performance? Does difference in data and training objectives play a part?**
>
> Both data and the training objective may play a part. Comparing DINOv1 and DINOv2 (same training objective), we can speculate that more training data leads to better probe performance. Comparing CLIP and Stable Diffusion (both trained on LAION), we can speculate that the SD training objective is superior to the CLIP one for the probe tasks.
>
> Additionally, we can also observe that there is a correlation between the performance of different 3D physical understanding tasks for the models we tested. For example, Stable Diffusion and DINOv2 are consistently the best for all tasks; CLIP and DINOv1 are consistently in the middle for all tasks; VQGAN is consistently the least performant among all the models for all tasks.
>
>
> **Q1: Curating the dataset: How are the regions selected? Are all regions kept in the final dataset? Is there any human intervention and selection?**
>
> (1) Regions are obtained as described in detail in Section 3.2 of the paper. Region pairs are randomly selected. Please see Lines 462-475 in the Appendix for more details.
>
> (2) No. Regions smaller than a threshold are not used, because regions that are too small are challenging even for humans to annotate.
>
> (3) In general there is no human intervention and selection. The exception is for ScanNetv2 where we have manually filtered out the annotations of poor quality. Although regions are automatically selected, we carry out a visual check of a subset to ensure the correctness of selected regions and region pairs.
>
>
> **Q2: Does region size affect task difficulty**
>
> We speculate that there is no correlation between the size of regions and difficulty of tasks. For example, material has larger regions because the annotations for the regions are generally larger but the performance of the probing is still not as good as other tasks with smaller regions, e.g., depth (comparison of average region size: 39171 pixels for material v.s. 9201 for depth). For each task, we have regions of different sizes, except for very small regions, as discussed in Lines 474-475 in the Appendix.
>
>
> **Q3: Does patch size affect the performance? What are the patch sizes of the evaluated models?**
>
> The patch sizes (patch side) for the evaluated models are: VQGAN 16 pixels; DINOv1 8 pixels; DINOv2 14 pixels; CLIP 14 pixels, while the performance for the tasks we studied is generally VQGAN < CLIP / DINOv1 < DINOv2. Therefore, there appears to be no correlation of the different model’s performance with the patch size.

---

> > ### Comment · Reviewer_4isv · 2024-08-11
> >
> > Thanks for the rebuttal and thanks for addressing my questions. I am happy to keep my rating.

---

### Official Review · Reviewer_Lz9q · 2024-07-11

**Soundness:** 3
**Presentation:** 3
**Contribution:** 3
**Rating:** 7
**Confidence:** 4

**Summary:**

In order to efficiently examine whether large vision models have explicit feature representations for different properties of the 3D physical scene, the paper proposes to linearly probe the features of different layers (and different time steps) from LVMs on specially designed binary classification problems. Extensive experiments demonstrate the effectiveness of the proposed probing scheme.

**Strengths:**

1. As far as I am concerned, the paper is the first to investigate the 3D knowledge learned by LVMs in a lightweight manner with linear probing.
2. For quality, the authors conduct extensive experiments on several datasets to validate the effectiveness of their strategy.
3. For clarity, the paper is well written and easy to follow.
4. For significance, it is crucial to investigate to what extent the pretrained LVMs understand the 3D physical world and the applications.

**Weaknesses:**

Although the paper unveils the 3D perception abilities of pretrained LVMs using linear probing, it is still very simple and far from real application in downstream tasks. It would be better if the author could give more downstream applications in addition to the one in the appendix.

**Questions:**

Please see the weakness section for details.

**Limitations:**

The limitations are well discussed in the paper.

---

> ### Author Rebuttal · Authors · 2024-08-06
>
> Thanks for your valuable comments. We have provided responses below:
>
> **W1: Downstream application of pre-trained LVMs**
>
> Concerning downstream applications, in the Appendix Section F, we showed that the features selected by the linear probe could be used for the task of normal prediction.
>
> We have also obtained preliminary results for the task of depth prediction using the features selected by the probe. In detail we use the Stable Diffusion feature from the specific time step and layer (Table 2 Row 7) selected by the linear probe for depth. We extract the image features using Stable Diffusion, and feed the features directly to a simple convolutional network to predict the depth. The network is trained on the NYUv2 Depth training dataset with the SD features frozen. For comparison we train the same convolutional network using image features from a frozen ResNet-50 pre-trained on ImageNet. The SD features have a substantially higher performance: Comparison between ResNet and SD features on NYUv2 Depth test dataset: $\delta_1$ 51.0 v.s. 80.8 (the higher the better); $\delta_2$ 81.5 v.s. 97.2 (the higher the better); $\delta_3$ 94.3 v.s. 99.6 (the higher the better); RMSE 1.0065 vs 0.5389 (the lower the better). This again illustrates the potential of the features selected by the probe for downstream applications. We will include this additional experiment in the Appendix.

---

### Author Rebuttal · Authors · 2024-08-06

We thank the reviewers for their appreciation of the importance of the problem and our approach, and for their comments.

Reviewer Lz9q:"The first to investigate 3D knowledge in LVMs in a lightweight manner with linear probing; extensive experiments; paper well-written; the problem is crucial to investigate";

Reviewer 4isv:"The work studies a significant question; important and of growing interest; novel benchmark for evaluating; writing is clear and easy to follow; thorough and organised experiments";

Reviewer Wno9:"The protocol considers various physical properties and is lightweight; overall structure and writing are easy to follow; the problem investigated is very interesting; new perspective and very valuable for future studies";

We have addressed the concerns raised by each reviewer separately under their respective reviews.

---

### Decision · Program_Chairs · 2024-09-25

**Decision:**

Accept (poster)

**Comment:**

This paper proposes to explore large vision models to determine to what extent they "understand" different physical properties of the 3D scene depicted in an image. The authors conduct extensive experiments with a variety of models and reach some encouraging conclusions. After a rebuttal discussion period, the paper received all positive reviews. The reviewers agreed that the proposed protocol is essential, the results are encouraging, and the paper is well written. AC recommends that the authors incorporate all reviewers' feedback into the final camera-ready version and congratulates the paper on its acceptance.